# Therapeutic Targeting of Inflammation and Virus Simultaneously Ameliorates Influenza Pneumonia and Protects from Morbidity and Mortality

**DOI:** 10.3390/v15020318

**Published:** 2023-01-23

**Authors:** Pratikshya Pandey, Zahrah Al Rumaih, Ma. Junaliah Tuazon Kels, Esther Ng, Rajendra Kc, Roslyn Malley, Geeta Chaudhri, Gunasegaran Karupiah

**Affiliations:** 1Viral Immunology and Immunopathology Group, Tasmanian School of Medicine, University of Tasmania, Hobart, TAS 7000, Australia; 2Infection and Immunity Group, Department of Immunology, The John Curtin School of Medical Research, Australian National University, Canberra, ACT 2601, Australia; 3Tasmanian School of Medicine, University of Tasmania, Hobart, TAS 7000, Australia

**Keywords:** influenza, viral pneumonia, dysregulated inflammatory cytokine response, combined oseltamivir and anti-inflammatory drug treatment, TNF/NF-κB and STAT3 signaling pathways

## Abstract

Influenza pneumonia is a severe complication caused by inflammation of the lungs following infection with seasonal and pandemic strains of influenza A virus (IAV), that can result in lung pathology, respiratory failure, and death. There is currently no treatment for severe disease and pneumonia caused by IAV. Antivirals are available but are only effective if treatment is initiated within 48 h of onset of symptoms. Influenza complications and mortality are often associated with high viral load and an excessive lung inflammatory cytokine response. Therefore, we simultaneously targeted the virus and inflammation. We used the antiviral oseltamivir and the anti-inflammatory drug etanercept to dampen TNF signaling after the onset of clinical signs to treat pneumonia in a mouse model of respiratory IAV infection. The combined treatment down-regulated the inflammatory cytokines *TNF*, *IL-1β*, *IL-6*, and *IL-12p40*, and the chemokines *CCL2*, *CCL5*, and *CXCL10*. Consequently, combined treatment with oseltamivir and a signal transducer and activator of transcription 3 (STAT3) inhibitor effectively reduced clinical disease and lung pathology. Combined treatment using etanercept or STAT3 inhibitor and oseltamivir dampened an overlapping set of cytokines. Thus, combined therapy targeting a specific cytokine or cytokine signaling pathway and an antiviral drug provide an effective treatment strategy for ameliorating IAV pneumonia. This approach might apply to treating pneumonia caused by severe acute respiratory syndrome coronavirus 2 (SARS-CoV-2).

## 1. Introduction

Pneumonia is a serious complication caused by inflammation of the lungs following infection with diverse viral pathogens, including seasonal and pandemic influenza A viruses (IAVs), that often result in lung pathology, respiratory failure, and death [1,2,3,4]. There are currently no treatments for viral pneumonia, including influenza pneumonia. Antivirals against IAVs are far from satisfactory if treatment is not initiated within 48 h of onset of disease symptoms [5]. Most individuals do not seek medical attention within this timeframe [6]. Thus, there is an urgent need to advance therapies that specifically treat severe IAV infection post-onset of symptoms.

A hyperactive immune response associated with dysregulated inflammatory cytokine/chemokine production, known as a ‘cytokine storm’ [7], causes pneumonia, lung pathology, and death [4]. Late after onset of symptoms (>48 h), the damaging effects of inflammatory cytokines and virus-mediated cytopathic effects together contribute to lung pathology, morbidity, and mortality [8]. An excessive early inflammatory cytokine/chemokine responses and leukocyte recruitment can be predictive of poor prognosis and poor clinical outcomes in IAV infections [1,9,10], and those inflammatory factors directly contribute to leukocyte recruitment into the lungs [8,11,12].

Some previous studies have evaluated the therapeutic potential of adjunctive anti-inflammatory drug interventions in severe respiratory diseases. Corticosteroids in combination with antiviral agents were shown to effectively alleviate the 2009 pandemic H1N1 influenza-associated pneumonia [13]. Similarly, Zheng et al. demonstrated amelioration of lung pathology and improved survival rates in H5N1 IAV-infected mice through co-administration of cyclooxygenase inhibitors and zanamivir [14]. Finally, Shi and colleagues reported that treatment with the anti-TNF drug, etanercept, at 2 h p.i. with H1N1 IAV protected mice from otherwise lethal infection [15].

We reasoned that the simultaneously targeting both the virus and inflammation late after the onset of disease signs would be an effective treatment strategy to ameliorate influenza pneumonia [16]. Targeting inflammatory cytokines or cytokine-signaling molecules will reduce inflammation and diminish leukocyte infiltration into the lung. Of the various cytokines implicated, tumor necrosis factor (TNF) is a crucial driver of inflammation in IAV-induced pneumonia [17,18,19,20]. Viral infection triggers rapid TNF production by the innate immune system through activation of the nuclear factor-kappa B (NF-κB) signaling pathway [21]. TNF exists in two distinct forms: soluble TNF (sTNF), and its precursor, transmembrane TNF (mTNF). Both forms of TNF can bind to their cognate receptors, TNF receptor type I and II (TNFRI and TNFRII), and mediate biological effects on various cell types, mainly through the activation of the NF-κB pathway [16,22]. Both TNFRs also exist as soluble (sTNFR) and membrane-bound (mTNFR) forms. The binding of sTNFRII or mTNFRII to mTNF can also transmit signals into mTNF-bearing cells and dampen inflammation through a process known as ‘reverse signaling’ [23,24].

Oseltamivir [25], an IAV neuraminidase (NA) inhibitor (NAI), is the most common anti-IAV drug currently used to treat individuals in high-risk groups [26]. It is only effective if treatment is initiated within 48 h of onset of disease symptoms [5,27]. Clinical trials and meta-analyses have shown that oseltamivir is ineffective in reducing severe disease phenotypes, including hospitalization and pneumonia, when treatment is commenced >48 h after the onset of symptoms [5].

We used a murine model of acute respiratory H1N1 IAV infection to investigate why oseltamivir is ineffective in reducing morbidity and mortality if treatment is initiated late after the onset of disease signs. We have found that oseltamivir can effectively reduce disease severity and mortality in IAV-infected mice even after the onset of disease signs but only when inflammation is also simultaneously targeted. We used etanercept, which is widely used to treat several inflammatory diseases [28], to reduce inflammation. Etanercept alone reduced clinical disease and lung pathology but did not protect mice from mortality. Similarly, treatment with only oseltamivir effectively reduced viral load, but animals died from severe lung pathology.

Dysregulated TNF production results in the dysregulation of specific cytokines, chemokines, and cytokine-signaling pathways, including the NF-κB and signal transducer and activator of transcription (STAT) 3 pathways [29,30]. We used an inhibitor of STAT3, S3I-201, as an alternative anti-inflammatory drug to reduce lung inflammation. Combined treatment with S3I-201 and oseltamivir reduced viral load, clinical illness, lung inflammation, and pathology in IAV-infected mice. The combined treatment with oseltamivir and etanercept or S3I-201 reduced the expression levels of an overlapping set of inflammatory cytokines and chemokines, many of which have been implicated in causing lung pathology during IAV infection.

Unlike a previous study [15], we have evaluated the efficacy of etanercept, combined with an antiviral drug, in the treatment of influenza pneumonia in a realistic timeframe that closely reflects timeframes when patients present at hospitals. Treating lung inflammation in humans (even before it has developed) from the day of infection is neither feasible nor practical for various logistical reasons. Additionally, targeting the STAT3 signaling pathway with an inhibitor combined with antiviral drug treatment to ameliorate influenza pneumonia is novel and has not been reported previously. Our findings not only help explain why NAIs are ineffective in treating severe disease and pneumonia caused by IAV infection if administered late after the onset of disease symptoms but also provide effective treatment strategies. Excessive levels of several cytokines and chemokines have been implicated in the pathogenesis of influenza pneumonia and lung pathology. Our results indicate that targeting only one inflammatory cytokine or a cytokine signaling pathway ameliorates lung pathology and protects mice from an otherwise lethal infection when combined with oseltamivir.

## 2. Materials and Methods

### 2.1. Animal Ethics Statement

Animal experiments were performed in accordance with protocols approved by the Animal Ethics Committee of the University of Tasmania (UTAS) (Protocol number A0016372) and the Animal Ethics and Experimentation Committee of the Australian National University (ANU) (Protocol numbers A2011/011 and A2014/018).

### 2.2. Mice

C57BL/6J wild-type (WT) female mice, bred under specific pathogen-free conditions, were obtained from the Australian Phenomics Facility (APF), ANU, Canberra, the Cambridge Farm Facility, UTAS, Tasmania or the Animal Resource Centre, Western Australia, Australia. In addition to WT mice, TNF deficient (TNF^−/−^) [31] and triple mutant (TM or mTNF^Δ/Δ^.TNFRI^−/−^.II^−/−^, expressing only mTNF but lacking sTNF and TNFRs) [30] mice, bred at the APF, ANU, were used in this study. One week prior to start of experiments, mice were transferred to the virus suite and allowed to acclimatize. Mice were used at 6–12 weeks of age.

### 2.3. Cell Lines and Viruses

Madin-Darby Canine Kidney (MDCK) cells (ATTC No. CCL-34) were cultured in Dulbecco’s Modified Eagle Medium (DMEM) supplemented with 2 mM l-glutamine (Sigma-Aldrich Pty Ltd., Macquarie, Australia), antibiotics (1× PSN; 50 U/mL penicillin, 50 μg/mL streptomycin, and 100 μg/mL neomycin) (Sigma-Aldrich Pty Ltd., Macquarie, Australia), and 10% heat inactivated fetal bovine serum (FBS). This will be referred to as cell growth medium. Cell cultures were maintained at 37 °C in a 5% CO_2_ atmosphere.

Stocks of Influenza A virus (IAV) H1N1 (A/PR/8/34) strain were propagated in 10-day old, specific-pathogen-free embryonated chicken eggs, and viral titers were determined in MDCK cells using plaque assay or median tissue culture infectious dose (TCID_50_) assay described elsewhere [32].

### 2.4. Virus Infection, Animal Weights, and Clinical Scores

Age-matched mice were anesthetized with isoflurane (UTAS) at 5% for induction and 2% for maintenance intranasally (i.n.) using the Stinger Streamline Rodent/Exotics Anesthetic Gas Machine (Advanced Anesthesia Specialists, Gladesville, New South Wales, Australia) or with tribromoethanol (ANU) at 160–240 mg/kg through the intraperitoneal (i.p.) route. Mice were infected with 3000 plaque forming unit (PFU) of IAV, housed in individually ventilated cages under biological safety level 2 containment facilities and monitored daily; weighed and scored for clinical signs of illness (scores ranged from 0 to 3 for each of the five clinical parameters, condition of hair coat, posture, breathing, lacrimation and nasal discharge, and activity and behavior) as described elsewhere [29,30]. For ethical reasons and as required by the animal ethics protocols, mice that were severely moribund with a clinical score of ≥10 and/or a body weight loss of ≥20% (UTAS) or 25% (ANU) were killed by CO_2_ asphyxiation, lung tissues collected for subsequent analyses and animals considered dead the following day.

### 2.5. Drug Treatments

Different groups of mice were administered with 100 µL of oseltamivir (Tamiflu; Roche Australia, Sydney, Australia) diluted in phosphate buffered saline (PBS) via oral gavage (o.g.) at 150 mg/kg (once daily) or 20 mg/kg (twice daily), etanercept (Enbrel; Pfizer Australia, Sydney, Australia) administered i.p. at 2.5 mg/kg diluted in PBS or S3I-201 (STAT3 inhibitor VI, Sigma-Aldrich Pty Ltd., Macquarie, Australia; cat. no. 573102) administered i.p. at 5 mg/kg after the onset of disease signs. S3I-201 was first dissolved in DMSO at 200 mg/mL and then further diluted in PBS to a working stock solution of 1 mg/mL.

### 2.6. Plaque Assay for Virus Quantification

Viral titers were determined as virus plaque forming units (PFU) per gram of lung tissue samples using a plaque assay as described elsewhere [32]. Briefly, homogenized lung samples were serially diluted (10-fold) and inoculated into the confluent MDCK monolayer in a 6-well tissue culture plate. After 1 h incubation, inoculum was removed, and the monolayer was covered with the agar overlay. After 4 days of incubation, the agar overlay was removed, and the cells were fixed with 10% formalin followed by staining with 0.1% crystal violet. Viral titers were calculated as PFU/g. For details, see Appendix A.

### 2.7. TCID_50_ Assay for Virus Quantification

The TCID_50_ assay for IAV quantification was previously described elsewhere [32]. Briefly, serial dilutions of lung homogenates were inoculated into the MDCK cell monolayer in a 96-well tissue culture plate. After 1 h of virus adsorption, inoculum was removed and the infected monolayer was incubated in the virus growth medium at 37 °C, 5% CO_2_ for 4 days. The cells were then fixed with 10% formalin and stained with 0.1% crystal violet to visualize virus induced cell cytopathic effect (CPE). We calculated the viral titer as TCID_50_/g using the Reed-Muench method [33]. For details, see Appendix A.

### 2.8. Lung Histopathological Examination

Lung tissue was sectioned and stained with Hematoxylin and Eosin (H&E) and assessed for histopathology using a semi-quantitative scoring system as described elsewhere [29,30]. For more details, see Appendix A.

### 2.9. RNA Extraction, cDNA Generation, and Quantitative Reverse Transcription Polymerase Chain Reaction (qRT-PCR)

RNA was extracted from the lung tissue homogenized in TRIzol solution (ThermoFisher Scientific, Melbourne, VIC, Australia; cat. no. 15596026), as described elsewhere [29,30], and cDNA was synthesized using RevertAid first strand cDNA synthesis kit (ThermoFisher Scientific, Melbourne, VIC, Australia; cat. No. K1622). PowerUp SYBR Green Master Mix (ThermoFisher Scientific, Melbourne, VIC, Australia; cat. No. A25742) was used to measure mRNA transcripts of cytokines/chemokines levels using quantitative real-time PCR (qRT-PCR). Recorded cycle threshold values were normalized to the housekeeping gene Ubiquitin C (UBC). Details on the procedure and primers used are presented in Appendix A.

### 2.10. Statistical Analysis

Statistical analyses of experimental data, as indicated in the legend to each figure, were performed using appropriate tests to compare results using GraphPad Prism 9 (GraphPad Software, Boston, MA, USA). A value of *p* < 0.05 was taken to be significant: *, *p* < 0.05; **, *p* < 0.01; ***, *p* < 0.001 and ****, *p* < 0.0001.

## 3. Results

### 3.1. Etanercept Improves Clinical Signs and Reduces Lung Pathology without Affecting IAV Load in WT and TM Mice but TNF Deficiency Exacerbates Lung Pathology

Etanercept dampens inflammation by reducing the availability of sTNF to signal via TNFR and through triggering reverse signaling via mTNF [34,35]. We first used wild-type (WT) mice and a triple mutant (TM) strain, engineered to express only the non-cleavable mTNF but not sTNF, TNFRI, or TNFRII [36], to establish that etanercept can reduce lung inflammation during an IAV infection. TM mice lack endogenous TNF signaling, like TNF^−/−^ mice, but respond to exogenous TNFR such as etanercept. WT and TM mice infected with 3000 PFU IAV i.n. were treated with 2.5 mg/kg etanercept or vehicle (PBS) intraperitoneally (i.p.) on days 1, 3, and 4 post-infection (p.i.). Animals were killed on day 5 p.i.

Compared with WT mice, mock-treated TM mice developed more severe disease as evident by higher losses in body weights and increased clinical scores, lung histopathological scores, pulmonary inflammatory cell infiltrates, oedema, and bronchial epithelial cell loss (Figure 1A–E). We generated lung histopathological scores from the microscopic examination of hematoxylin and eosin (H&E)-stained lung sections using a semi-quantitative scoring system described elsewhere [29,30]. Treatment with etanercept significantly reduced weight loss, clinical scores, and histopathological scores in both strains of mice (Figure 1A–E) but did not affect viral load (Figure 1F). Microscopic examination of lung histological sections revealed significant reductions in parenchymal edema and inflammatory cell infiltration by etanercept treatment, consistent with reduced weight loss, clinical scores, and histopathological scores (Figure 1G).

We tested the response of TNF^−/−^ mice to IAV infection as we wanted to use them as controls in experiments with etanercept and oseltamivir combined treatment. We found that IAV-infected TNF^−/−^ mice had significantly higher clinical scores than WT mice, evident from days 4–6 p.i. although both strains had comparable body weight losses from days 2–6 p.i. (Appendix A). The lung viral load was comparable in both strains of mice (Appendix A). Compared to WT mice, TNF^−/−^ mice had significantly higher histopathological scores (Appendix A). Expectedly, there were no pathological changes in lung sections of mock-infected WT (Appendix A) and TNF^−/−^ (Appendix A) mice. Pathological changes were evident in IAV-infected WT mice (Appendix A) and were more pronounced TNF^−/−^ mice (Appendix A), consistent with a previous report [37].

### 3.2. Etanercept Combined with a Standard Dose of Oseltamivir (40 mg/kg) Daily Treatment Reduces Morbidity and Lung Pathology but Has No Effect on Viral Load

Etanercept treatment beginning at day 1 p.i. with IAV improved clinical disease and lung pathology in mice (Figure 1). However, most patients seek medical attention late after the onset of symptoms [6]. Furthermore, the cytopathic effects of exponentially increasing viral load also contribute to lung injury at that stage of the disease, but etanercept does not affect lung viral load. We reasoned that combined treatment with etanercept and oseltamivir simultaneously would be necessary to minimize disease severity effectively.

An oral dose of 150 mg oseltamivir twice daily is well-tolerated in adult humans [38] and is equivalent to a dose of 20 mg/kg in mice administered twice daily [39]. Mice infected with a lethal dose of 3000 PFU IAV i.n. were given an oral dose of oseltamivir at 20 mg/kg (twice daily; total 40 mg/kg/day), one dose of etanercept, or both drugs (combined) on day 3 p.i. after the onset of disease signs. Additional doses of oseltamivir were given on days 4 and 5 p.i. All mice were killed on day 6 p.i. Since the clinical course of IAV disease is short in mice, it leaves a very narrow window for treatment. We commenced treatment from day 3 p.i, so that animals receive appropriate treatment(s) for at least 2 days before they succumb to infection.

Oseltamivir or combined treatment significantly reduced weight loss, clinical scores, and lung histopathological scores (Appendix A), but did not affect lung viral load (Appendix A). A single dose of etanercept also significantly reduced clinical scores but had no impact on weight loss or histopathological scores (Appendix A).

Oseltamivir or etanercept significantly reduced the levels of mRNA transcripts for TNF, IL-6, IL-12p40, CCL2, and CCL5 (Appendix A). In addition to these cytokines and chemokines, the combined treatment significantly downregulated levels of expression of IL-1β and CXCL10 (Appendix A). The results from these experiments suggested a higher dose of oseltamivir would be required to effectively control virus replication [40,41].

### 3.3. Combined Treatment with Etanercept and High Dose Oseltamivir Reduces Morbidity, Lung Viral Load, and Pathology in IAV-Infected Mice

A standard dose of 20 mg/kg oseltamivir (administered twice daily) was ineffective in reducing viral load. Furthermore, a single dose of etanercept had no impact on weight loss or histopathological scores. Chronic inflammatory diseases like rheumatoid arthritis are treated with etanercept once or twice a week [28]. However, the levels of TNF produced during an acute viral infection are likely to be different to those during a chronic inflammatory condition. Therefore, we administered etanercept and high dose oseltamivir (150 mg/kg) once daily in subsequent experiments, unless indicated otherwise.

Groups of IAV-infected WT, TM, and TNF^−/−^ mice were treated with oseltamivir, etanercept, or both drugs after onset of disease signs on day 3 p.i. Treatment was continued on days 4 and 5 p.i. once daily and animals were killed at day 6 p.i. TNF^−/−^ mice were used as negative controls as they do not respond to etanercept treatment [36]. TM mice lack endogenous TNF signaling but respond to etanercept treatment (Figure 1).

Mock-, etanercept-, or oseltamivir-treated WT mice continued to lose weight until day 6 p.i. when animals were killed for various analyses. In contrast, WT mice given the combined treatment stopped losing weight just one day after initiation of treatment and by day 6 p.i.; body weight loss in this group was significantly lower than the other groups (Figure 2A). All treatment regimens significantly lowered clinical scores from day 4 p.i., but the most significant reduction was in the combined treatment group (Figure 2B). Lung viral loads were similar in mock and etanercept treated groups but significantly reduced by oseltamivir or combined treatment compared with the mock-treated group (Figure 2C). Notably, once-daily treatment with 150 mg/kg oseltamivir alone effectively reduced viral load late after the onset of disease signs. There were significant reductions in lung histopathological scores in etanercept- but not oseltamivir-treated animals compared to mock-treated mice, but the combined treatment showed the biggest reduction (Figure 2D).

In IAV-infected TNF^−/−^ mice, only the combined treatment reduced weight loss (Figure 2E). Mice given oseltamivir or the combined treatment had significantly lower clinical scores than etanercept- or mock-treated animals (Figure 2F). Lung viral load and histopathological scores were also reduced by oseltamivir or combined treatment but not by etanercept or mock treatment (Figure 2G,H). The reduced viral load likely contributed to the lower clinical and histopathological scores in TNF^−/−^ mice.

TM mice exhibited beneficial effects from the single or combined treatment regimens, similar to observations made in WT mice (Figure 2I–L). On day 6 p.i., etanercept treatment reduced weight loss compared to mock treatment, but the effect was more pronounced in oseltamivir or combined treatment groups (Figure 2I). All treatment regimens significantly reduced clinical scores, with the combined treatment having the most prominent effect (Figure 2J). Oseltamivir or combined treatment, but not etanercept, reduced lung viral load (Figure 2K). Lung histopathological scores were significantly reduced by oseltamivir or etanercept treatment, but the combined treatment reduced it to a substantially greater extent (Figure 2L).

### 3.4. Combined Treatment with Etanercept and High Dose Oseltamivir Reduces IAV Infection-Induced Lung Pathology Effectively in WT and TM Mice

Microscopic examination of lung histological sections revealed dramatic reductions in parenchymal edema and damage to bronchial and alveolar walls in WT mice given the combined treatment compared to the other treatment groups (Figure 3A–H). Focal leukocyte infiltration was most abundant in lungs of WT mice given mock-treatment compared with oseltamivir- or etanercept-treated groups, but it was only moderate in mice given the combined therapy (Figure 3A–H). In TNF^−/−^ mice, IAV-induced lung edema and inflammatory cell recruitment were reduced by oseltamivir or combined treatment, but not by etanercept (Figure 3I–P). However, the extent of improvements was less compared to WT mice. All treatment regimens ameliorated edema and inflammatory cell infiltration in TM mice, except for mock treatment, but the degree of improvement was highest in the combined-treated group (Figure 3Q–X).

### 3.5. Combined Daily Treatment with Etanercept and High Dose Oseltamivir Reduces Expression of Inflammatory Cytokines and Chemokines

To obtain an insight into the mechanisms through which treatment with etanercept combined with a higher dose of oseltamivir, both administered daily (Figure 2 and Figure 3), we focused on the effects of treatment on inflammatory cytokine and chemokine mRNA transcript levels using lung tissue samples from the experiment described in Figure 2 and Figure 3.

Compared to mock treatment, etanercept or oseltamivir reduced mRNA transcripts for *TNF*, *IL-1β*, and *IL-12p40* (Figure 4A,B,D), whereas the combined treatment was more effective in decreasing the levels of those cytokines as well as *IL-6* and the chemokines CCL2, CCL5, and *CXCL10* (Figure 4C,E–G).

Taken together, the effectiveness of the combined treatment regimen in reducing weight loss, clinical disease, lung pathology, and viral load was associated with significant reductions in *TNF*, *IL-1β*, *IL-6*, *IL-12p40*, *CCL2*, *CCL5*, and *CXCL10*. It was evident that the higher dose of oseltamivir resulted in significant reductions in viral load and levels of *IL-1β* more effectively (Figure 4) compared to treatment with a lower dose oseltamivir (Appendix A).

### 3.6. Combined Daily Treatment with Etanercept and High Dose Oseltamivir Protects Mice from Lethal IAV Infection

We determined whether the higher dose of oseltamivir administration in the combined treatment regimen afforded protection against influenza pneumonia and lethal disease. IAV-infected mice were treated with high dose oseltamivir, etanercept, or both drugs after the onset of disease signs at day 3 p.i. once daily for up to 20 days. Animals were monitored for morbidity and mortality until day 21 p.i. when all surviving animals were killed. TNF^−/−^ mice, which do not respond to etanercept treatment (Figure 2) were infected and treated similarly for use as controls.

Both WT and TNF^−/−^ mice infected with IAV continued to lose weight from day 2 p.i. and exhibited clinical signs of disease from day 3 p.i. (Figure 5A–D). All mock-treated WT mice succumbed to infection and were killed for ethical reasons by day 6 p.i. Etanercept treatment prolonged survival by 1 day in 4 of 5 (80%) IAV-infected WT mice (Figure 5E). Oseltamivir treatment alone protected 1 of 5 (20%) mice from lethal IAV infection while the remaining animals succumbed on days 6 and 7 p.i. Mice given the combined treatment had the highest survival rate wherein one mouse succumbed to the infection on day 11 p.i., but 80% of animals survived until day 21 when they were killed (Figure 5E). Mock-treated TNF^−/−^ mice succumbed to the infection on days 5 or 6 p.i. (Figure 5F), and there were no beneficial effects of etanercept, oseltamivir, or the combined therapies as all IAV-infected animals succumbed by day 7 p.i.

Data presented in the preceding section indicated that viral load in WT mice was significantly reduced by oseltamivir alone or combined treatment regimens (Figure 2C). However, a greater proportion of mice in the combined treatment group survived (Figure 5E). Thus, although high dose oseltamivir was effective in reducing viral load late after the onset of disease signs, it alone was insufficient to protect against influenza pneumonia, morbidity, or mortality. Therefore, both virus and inflammation must be targeted simultaneously to afford protection against influenza pneumonia.

### 3.7. STAT3 Inhibitor in Combination with Oseltamivir Reduces Lung Viral Load and Improves Lung Pathology and Morbidity Associated with Severe IAV Infection

The STAT3 cytokine signaling pathway is downstream of the NF-κB pathway, and dysregulated TNF levels cause hyperactivation of STAT3, which also correlated with severe pneumonia during respiratory ectromelia virus (ECTV) infection [29,30]. ECTV causes mousepox, a surrogate mouse model for smallpox caused by the variola virus in humans [42]. Treatment of ECTV-infected mice with a selective STAT3 inhibitor, SI-301, significantly reduced lung pathology, but it was insufficient to protect the animals [29]. However, inhibition of STAT3 combined with an antiviral drug, cidofovir, significantly reduced clinical disease and viral load and ameliorated lung pathology [36]. The reduced morbidity was associated with reductions in inflammatory cytokine/chemokine gene/protein expression. We hypothesized that a similar approach of simultaneous targeting of the virus and the STAT3 signaling pathway might also be effective in the IAV pneumonia model.

We investigated whether the combined treatment regimen with S3I-201 and high dose oseltamivir would be effective in reducing morbidity. Groups of IAV infected mice were treated with 150 mg/kg oseltamivir, 5 mg/kg S3I-201, or both drugs (combined) on days 3 and 4 p.i. after the onset of disease signs. Since one of the mock-treated animals was severely moribund, all mice were killed on day 5 p.i. for ethical reasons.

High dose oseltamivir did not affect weight loss and modestly reduced lung histopathological scores but had a significant impact in reducing clinical scores and lung viral load (Figure 6A–D). S3I-201 or combined treatment significantly reduced weight loss, clinical scores, and lung histopathological scores, while the latter also reduced the lung viral load and was by far the most effective in reducing all parameters measured (Figure 6A–D). Compared to other treatment groups, the combined treatment diminished lung inflammation and damage, evident by reduced edema, cellular infiltration, and bronchial epithelial necrosis (Figure 6E–L).

Through lowering the lung viral load, oseltamivir significantly reduced the levels of expression of *TNF*, *IL-1β*, *IL-12p40*, and *CCL5* (Figure 6M,N,P,R). The combined treatment effectively dampened lung inflammation through reductions in the mRNA levels of *TNF*, *IL-1β*, *IL-6*, *IL-12p40*, *CCL2*, *CCL5*, and *CXCL10* (Figure 6M–S).

Taken together, simultaneous targeting of the virus and host inflammatory response using etanercept or a STAT3 inhibitor is an effective strategy to treat severe IAV pneumonia, particularly when treatment needs to be initiated late after the onset of signs and symptoms.

## 4. Discussion

Pneumonia is a severe complication caused by inflammation of the lungs due to infection with diverse viral pathogens that often results in respiratory failure and death. Seasonal and pandemic influenza viruses [1,2,3,4], variola virus (agent of smallpox) [43], and severe acute respiratory syndrome coronavirus 2 (SARS-CoV-2) [44] are leading examples. Pneumonia is one of the most common and life-threatening complications of IAV infection [2]. Globally, nearly 1 billion people are infected with seasonal influenza annually, with 3–5 million cases of severe illness and 300,000–650,000 deaths [45,46]. In 2017 alone, 145,000 deaths and about 9.5 million hospitalizations were attributed to influenza-associated lower respiratory tract infection, including pneumonia [47].

There are no specific treatments for influenza pneumonia other than hospitalization and supportive care. Antivirals against IAV are ineffective against severe influenza and pneumonia if treatment is not initiated within 48 h of disease symptoms. Most individuals do not seek medical attention within this timeframe [6]. There is, therefore, an urgent need to advance therapies that specifically treat severe influenza post-onset of symptoms. This study has addressed two different, but critical questions related to influenza pneumonia. The first relates to the development of an effective treatment for IAV pneumonia for which specific treatments are currently not available. The second pertains to why antivirals against IAV are ineffective in reducing morbidity and mortality if treatment is not initiated within 48 h post-onset of disease symptoms.

Strategies that target the virus alone with antivirals have shown limited clinical efficacy in treating IAV infection, especially when treatment is initiated late during illness. Oseltamivir is only effective in reducing morbidity, hospital admissions, or disease complications when treatment is commenced within 48 h of onset of symptoms [5]. The ineffectiveness of antivirals in ameliorating pneumonia or reducing morbidity late after onset of disease symptoms is not unique to IAV infection. Several studies, including our own, have shown that antiviral agents including cidofovir or tecovirimat (Tpoxx) are effective in treating Orthopoxvirus (OPXV) infections, but are only partially protective if administered late after disease onset [36,48,49,50]. Similarly, antiviral drugs for the herpes simplex virus (acyclovir and valacyclovir) are effective only when administered within 72 h of lesion appearance [51]. In patients hospitalized with SARS-CoV-2 infection, antiviral drugs, including remdesivir, lopinavir, and interferon, were found to have little or no effect in reducing disease progression and mortality [52].

On the other hand, although strategies to limit or dampen inflammation during IAV infection have shown potential benefits, evidence on clinical benefits from the use of only anti-inflammatory drugs is inconclusive, particularly in hospitalized patients or when the treatment initiation is delayed [17]. Corticosteroids, widely used as anti-inflammatory agents, were ineffective in preventing mortality of IAV H5N1 infected mice when treatment was initiated late after the onset of disease signs [53,54]. Other important regulators of inflammation, including peroxisome proliferator-activated receptor agonist and cyclooxygenase inhibitors, showed no survival benefits in H5N1 IAV infected mice when administered 48 h p.i. [14]. In contrast, immunomodulatory agents targeting sphingosine-1-phosphate receptor agonist [55], C-C chemokine receptor 2 [56], or AMP-activated protein kinase [57] protected mice against a lethal influenza viral challenge. These agents were administered as prophylactics or very early after virus inoculation. Such treatment regimens have a minimal translatory application, especially since patients often seek medical advice after the onset of symptoms or once they have developed pneumonia.

Both high viral load and high levels of inflammatory factors (cytokines, chemokines, and transcription factors) increase the risk of illness severity and mortality after the onset of disease signs in H1N1 or H5N1 IAV infections in mice and humans [1,9,58,59]. We used that as a rationale for the simultaneous targeting of virus replication and the host inflammatory response as an approach to reduce morbidity and mortality in IAV-infected mice. We co-administered oseltamivir and anti-inflammatory drugs (etanercept or SI-301) to reduce lung viral load and pathology and improve survival rates in mice with H1N1 influenza pneumonia. We first focused on TNF, which is produced in the early phases of IAV infection and is associated with illness severity and morbidity in mice [60], swine [61], and human [62] IAV infections. Anti-TNF drugs are used widely to treat immune-mediated inflammatory diseases, including rheumatoid arthritis, Crohn’s disease, and psoriatic arthritis [63]. However, in contrast to treating chronic inflammatory diseases, our results show that the effective treatment of IAV pneumonia would require multiple doses of etanercept, indicating that the levels of TNF produced in the lung during acute respiratory viral infection might be different from those produced during chronic inflammatory disease and as a result, requires multiple doses of etanercept.

Combined treatment with etanercept and oseltamivir late after the onset of disease signs reduced morbidity, viral load, and lung pathology in IAV-infected mice through downregulation of inflammatory factors. These included *TNF*, *IL-1β*, *IL-6*, *IL-12p40*, *CCL2*, *CCL5*, and *CXCL10*. Treatment with etanercept or oseltamivir alone had limited clinical benefits given that they reduced the mRNA levels of some inflammatory factors but not to the extent of the combined treatment regimen. In a previous study, Shi and colleagues reported that etanercept treatment 2 h p.i. with H1N1 IAV protected mice from otherwise lethal infection [15]. Our study evaluated the efficacy of combined therapy in a realistic timeframe that closely reflects timeframes when patients present at hospitals. Treating lung inflammation in humans, even before it has developed, from the day of infection is neither feasible nor practical. However, if exposure to virulent or pandemic strains of IAV is identified very early, then anti-IAV drugs would effectively minimize the risk of severe disease, morbidity, and mortality [25]. We found that a higher dose of oseltamivir at 150 mg/kg/day, but not a standard mouse dose of 40 mg/kg/day [64], effectively reduced the lung IAV load. The reason for that may be related to the lethal dose of IAV that was used in our study.

We made a similar finding when the STAT3 signaling pathway was targeted instead of the TNF/NF-κB pathway to reduce lung inflammation. Combined treatment with S3I-201 and oseltamivir reduced viral load, disease signs, and lung pathology in IAV-infected mice through the downregulation of an overlapping set of cytokines and chemokines as combined treatment with etanercept and oseltamivir, i.e., *TNF*, *IL-1β*, *IL-6*, *IL-12p40*, *CCL2*, *CCL5*, and *CXCL10*. These cytokines and chemokines enhance acute phase signaling; recruit inflammatory cells, including neutrophils, monocytes, and T lymphocytes to the site of infection; and trigger secondary cytokine production, resulting in lung inflammation and pathology [7]. Thus, blockading cytokine(s) or cytokine signaling pathways combined with antiviral treatment will be expected to reduce leukocyte recruitment into the lung. Indeed, the combined treatment using either etanercept or SI-301 significantly reduced leukocyte migration to the lungs as evident histologically. In ECTV-infected mice, combined treatment with etanercept and cidofovir reduced recruitment of inflammatory monocytes to the lung [36]. Inflammatory monocytes produce cytokines like *IL-1*, *TNF*, *IL-6*, *CCL2*, and *CXCL10*.

Various stimuli, including viral infection and inflammatory cytokines, activate the NF-κB and STAT3 signaling pathways. TNF and IL-1 activate NF-κB [65], which enhances cytokine expression, including IL-6, which is a potent inducer of STAT3 activation [16,66,67]. NF-κB and STAT3 cooperatively regulate the expression of several inflammatory cytokines, such as IL-6, CCL2, CCL5, IL-8, and IL-17 [68,69]. A recent study evaluating global chromatin binding has revealed more than 36,000 *cis*-regulatory regions that can potentially bind to both STAT3 and NF-κB [70]. NF-κB and STAT3 can collaboratively induce their target gene expression through direct physiological interaction or cooperative binding at a subset of gene promoters/enhancers [68,70,71].

Besides IAV pneumonia, respiratory OPXV [29] and SARS-CoV-2 pneumonia [72,73] are also associated with high viral load and dysregulated inflammation and may benefit from combined antiviral and anti-inflammatory treatment approaches. We have recently shown therapeutic efficacy of combined cidofovir (a viral DNA polymerase inhibitor) and etanercept or STAT3 inhibitor treatment in ameliorating lung pathology and protecting mice from lethal OPXV pneumonia [36]. We believe that STAT3 inhibition will be more appropriate in individuals who cannot be treated with etanercept due to contraindications to the drug or when TNF may not be the driver of lung inflammation. Indeed, in a model of respiratory OPXV infection in TNF^−/−^ mice, combined treatment with cidofovir and SI-301 effectively reduced viral load and lung pathology [36].

In summary, combined treatment targeting virus and TNF/NF-κB or STAT3 pathways reduces viral load, clinical illness, and lung pathology in IAV-infected mice through downregulation of inflammatory cytokines and chemokines, many of which are implicated in disease severity and lung pathology caused by other respiratory viruses, including OPXV [36] and coronaviruses [74]. Therefore, the focus of clinical management of patients with severe viral pneumonia, associated with high viral load and dysregulated inflammation, should be on effective control of both viral replication and inflammatory cytokine and chemokine responses by the combined antiviral and anti-inflammatory drug treatment approach.

## Figures and Tables

**Figure 1 viruses-15-00318-f001:**
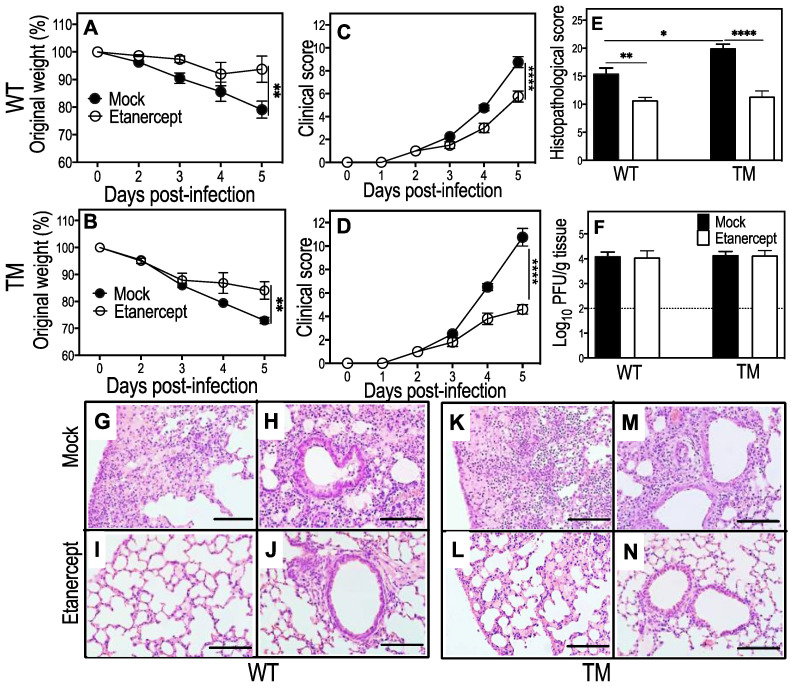
Etanercept treatment reduces weight loss, clinical scores, and lung pathology but not viral load in IAV-infected WT and TM mice. Age-matched groups of female WT and TM mice (*n* = 4 or 5) were infected with 3000 PFU IAV i.n. Animals were treated with 2.5 mg/kg etanercept or diluent (mock) on days 1, 3, and 4 p.i. Animals were killed on day 5 p.i. and lungs collected for analyses. Weight loss (**A**,**B**) and clinical scores (**C**,**D**) were analyzed using two-way ANOVA with Sidak’s post-tests and expressed as means ± SEM. Viral load data (**E**) were log-transformed, analyzed using ordinary one-way ANOVA test with Fisher’s least significant difference (LSD) tests, and expressed as means ± SEM. Histopathological scores (**F**), based on microscopic examination of lung histology H&E sections (**G**), were examined using bright field microscope on all fields at 400× magnification. Histopathological scores were analyzed by ordinary one-way ANOVA test with Tukey’s post-tests and expressed as means ± SEM. Lung histology sections (**G**–**N**) show reductions in edema, leukocyte infiltration, and damage to alveolar septa in lungs of IAV-infected WT and TM mice by etanercept treatment. *, *p* < 0.05; **, *p* < 0.01, and ****, *p* < 0.0001. The. Broken line in panel E corresponds to the limit of virus detection. Bars in panel G-N correspond to 100 μm. TM, triple mutant mice that express mTNF but not sTNF, TNFRI, or TNFRII; WT, wild-type mice.

**Figure 2 viruses-15-00318-f002:**
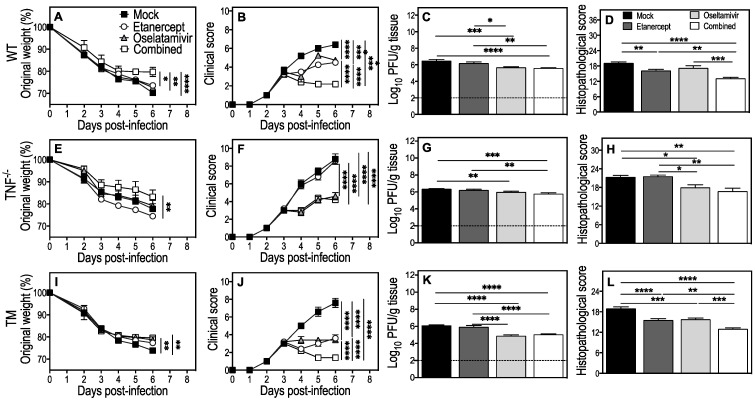
Combined treatment with etanercept and oseltamivir reduces clinical scores, lung viral load, and pathology in IAV-infected WT, TNF^−^/^−^, and TM mice. Age-matched groups of WT, TNF^−^/^−^, and TM (*n* = 4 or 5) female mice were infected with 3000 PFU IAV i.n. and treated with oseltamivir (150 mg/kg), etanercept (2.5 mg/kg), or a combination (combined) on days 3, 4, and 5 p.i. Animals were monitored for weight loss (**A**,**E**,**I**) and clinical scores (**B**,**F**,**J**) until day 6 p.i., when all animals were euthanized and lung tissue collected for various analyses. Data were analyzed by two-way ANOVA with Sidak’s post-tests and expressed as means ± SEM. Viral load (**C**,**G**,**K**) data was log-transformed, analyzed using ordinary one-way ANOVA test followed by Fisher’s LSD tests, and expressed as means ± SEM. Histopathological scores (**D**,**H**,**L**) were derived from microscopic examination of lung histology H&E sections (presented in Figure 3), analyzed using ordinary one-way ANOVA test followed by Tukey’s multiple comparisons tests and expressed as means ± SEM. *, *p* < 0.05; **, *p* < 0.01; ***, *p* < 0.001, and ****, *p* < 0.0001. Broken lines in panels (**C**,**G**,**K**) correspond to the limit of virus detection.

**Figure 3 viruses-15-00318-f003:**
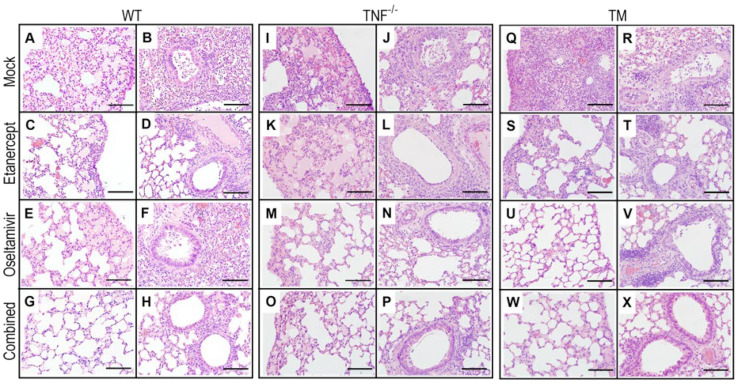
Combined daily treatment with etanercept and high dose oseltamivir reduces IAV infection-induced lung pathology in WT and TM mice. Lung tissue sections were obtained from mice that were infected and treated as described in Figure 2. Briefly, groups of WT, TNF^−^/^−^, and TM mice (*n* = 4 or 5) were infected with 3000 PFU IAV i.n. and then treated with oseltamivir, etanercept, or both drugs combined on days 3, 4, and 5 p.i. Animals were killed on day 6 p.i., lungs were collected, fixed in 10% neutral buffered formalin, processed, embedded in paraffin blocks, sectioned, stained with H&E, and examined using bright field microscope on all fields at 400× magnification. Bars in panels (**A**–**X**) correspond to 100 μm.

**Figure 4 viruses-15-00318-f004:**
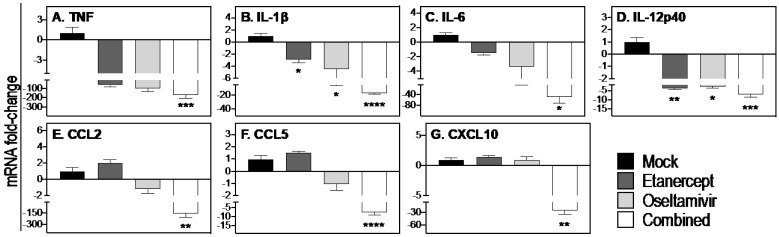
Combined daily treatment with etanercept and high dose oseltamivir reduces expression of inflammatory cytokines and chemokines. Lung tissues were obtained from WT mice that were infected and treated as described in Figure 2. Briefly, WT (*n* = 4 or 5) mice were infected with 3000 PFU IAV i.n. and treated with oseltamivir (150 mg/kg), etanercept, or combined treatment on days 3, 4, and 5 p.i. Animals were killed on day 6 p.i. and lungs collected for quantifying levels of expression of selected cytokines and chemokines using qPCR (**A**–**G**). Data were analyzed by one-way ANOVA with Holm-Sidak’s multiple comparisons tests and expressed as mean fold-change relative to the mock treated group ± SEM. *, *p* < 0.05; **, *p* < 0.01; ***, *p* < 0.001, and ****, *p* < 0.0001.

**Figure 5 viruses-15-00318-f005:**
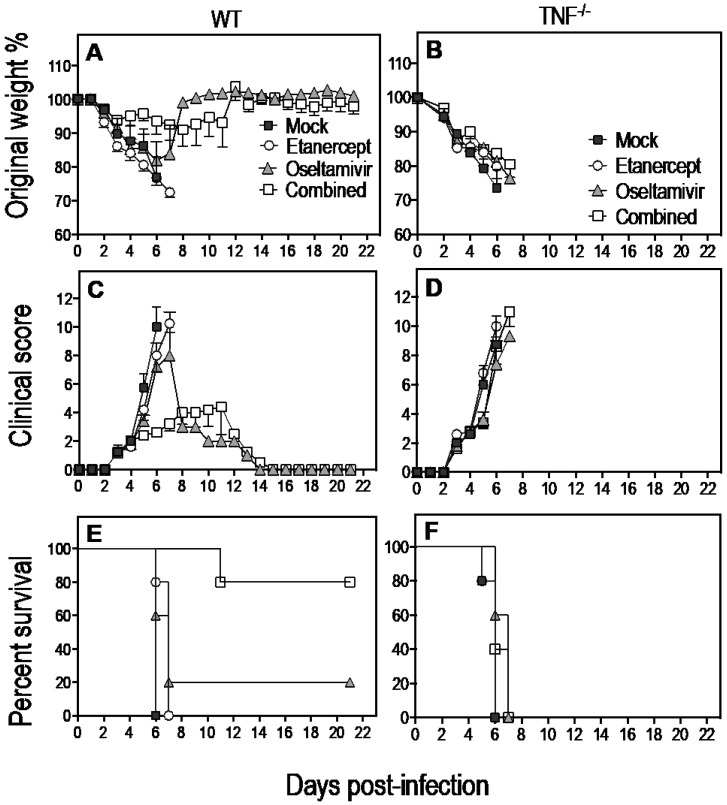
Combined daily treatment with etanercept and high dose oseltamivir reduces weight loss and clinical scores and improves the survival rate of WT but not TNF^−^/^−^ mice infected with IAV. Age-matched groups of WT and TNF^−^/^−^ (*n* = 4 or 5) mice were infected with 3000 PFU IAV i.n. Animals were treated with oseltamivir or etanercept or both drugs (combined) from day 3 p.i and treatment continued until day 20 p.i. Animals were monitored daily for weight loss (**A**,**B**), clinical scores (**C**,**D**), and survival (**E**,**F**). All mock-treated WT mice died by day 6 p.i., whereas etanercept-treated animals succumbed between days 6–7 (**E**). Four of five oseltamivir-treated WT mice succumbed between days 6–7 p.i. whereas one animal was alive at day 21 p.i. The combined treatment resulted in 80% of WT mice surviving at day 21 (**E**). The median survival time for mice treated with etanercept or oseltamivir was 7 days, compared with a median survival of 6 days for mock-treated mice (log-rank test, mock vs. etanercept *p* = 0.0237; mock vs. oseltamivir, *p* = 0.0736). Survival for combined treated animals was 80% at the last time point (day 21 p.i.), hence the median survival time was >21 days (log-rank test, *p* = 0.0047 relative to mock-treated mice). Combined treatment significantly increased median survival compared to etanercept (*p* = 0.0035) or oseltamivir (*p* = 0.0358) treatments. Mock-treated TNF^−^/^−^ mice succumbed to infection on days 5–6 p.i. (**F**) and there were no beneficial effects of the different treatment regimens as all IAV-infected mice succumbed by day 7 p.i. Weight loss (**A**,**B**) and clinical scores (**C**,**D**) data were analyzed using two-way ANOVA with Sidak’s post-tests and expressed as means ± SEM. Survival data (**E**) were analyzed by Log-rank (Mantel-Cox) test.

**Figure 6 viruses-15-00318-f006:**
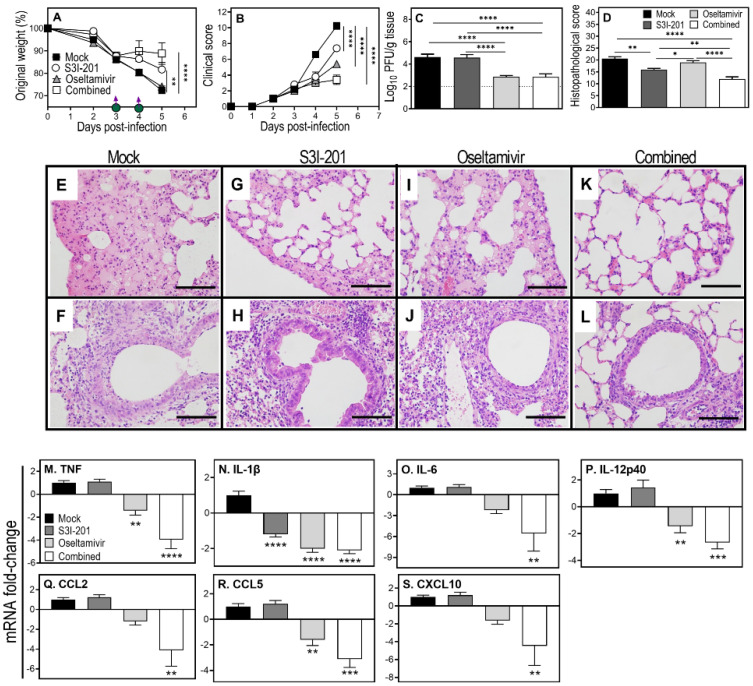
Combined treatment with high dose oseltamivir and S3I-201 reduces clinical scores, lung viral load, pathology, and downregulates mRNA transcripts for pro-inflammatory cytokines and chemokines in IAV-infected mice. Age-matched groups (*n* = 5) of female WT mice were infected i.n. with 3000 PFU IAV and treated with S3I-201 (5 mg/kg), oseltamivir (150 mg/kg), or both drugs (combined) on days 3 and 4 p.i. For ethical reasons, animals were killed on day 5 p.i. and lung tissue collected for various analyses. Weight loss (**A**) and clinical scores (**B**) were monitored until day 5 p.i. Viral load (**C**) data was log-transformed and analyzed using ordinary one-way ANOVA with Fisher’s LSD post-tests. Histopathological scores (**D**) were derived from microscopic examination of the lung histology H&E sections using bright field microscope on all fields at 400× magnification, presented in panel (**E**–**L**). Data are expressed as means ± SEM and were analyzed using two-way ANOVA (**A**,**B**) with Tukey’s multiple comparisons tests (**A**) and Dunnett’s multiple comparisons tests (**B**), or an ordinary one-way ANOVA followed by Tukey’s multiple comparisons tests (**D**). mRNA transcript analysis was performed in the separate experiment using the similar treatment strategy as described above. Gene expression levels of the indicated cytokines and chemokines were quantified using qPCR (**M**–**S**). Data are expressed as mean fold-change relative to the mock treated group ± SEM and were analyzed using one-way ANOVA with Holm-Sidak’s multiple comparisons tests. *, *p* < 0.05; **, *p* < 0.01; ***, *p* < 0.001, and ****, *p* < 0.0001. Broken line in panel C corresponds to the limit of virus detection. Bars in panels (**E**–**L**) correspond to 100 μm.

## Data Availability

All data presented in this study are available within the Appendix A associated with the article. Any additional information may be requested from the corresponding author.

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
