# Peer review of "Therapeutic Targeting of Inflammation and Virus Simultaneously Ameliorates Influenza Pneumonia and Protects from Morbidity and Mortality"

_viruses, 2023, doi:10.3390/v15020318_

Round 1
Reviewer 1 Report
The work by Pandey and colleagues entitled “Therapeutic targeting of inflammation and virus simultaneously ameliorates influenza pneumonia and protects from morbidity and mortality” is based on a mouse model of influenza virus infection. It mimics cases of acute illness by inoculating animals with an H1N1 virus.
A rational analysis of the existing data on this pathology allowed the authors to develop a treatment strategy targeting both the virus and the host immune response, which is detrimental in this acute syndrome.
The authors seem to present this approach as innovative, yet this strategy was already considered more than 20 years ago. I therefore think that the authors should include a paragraph in the "introduction" section specifying the novelty that their work brings to the subject. The targeting of STAT3 is however new and original. The results are very impressive, especially if one considers the survival of the animals with this treatment. The use of mice invalidated for different forms of TNF is very relevant and adds much to the demonstration.
Overall, the data presented are relevant and conclusive. However, I believe that the qPCR data in Fig. 4 and 6 should be presented differently, as it is not possible to assess the level of cytokine induction in the infected and untreated condition. The authors should change the reference and use an uninfected control for example.
My feeling is that the manuscript needs to be reworked (without addition of new experiments) in order to meet the publication requirements of “Viruses”.
Minor comments :
Abstract and introduction: the first sentences must be modified since influenza is not the only cause of pneumonia.
Line 43: “An over-exuberant immune response…” please find a more appropriate term to qualify the cytokine storm observed during severe influenza pathology.
Line 180: change “Clinical Disease” for “Clinical Symptoms”
Line 182: “Etanercept mediates anti-inflammatory effects by neutralizing sTNF and…” to my knowledge, Etanercept does not neutralize sTNF but compete with its receptor.
Line 320: “…dramatic reductions…” as this event is beneficial to the infected host, I think the authors should use another term than "dramatic".
Author Response
We thank Reviewer 1 for their positive and constructive comments on our manuscript, "Therapeutic targeting of inflammation and virus simultaneously ameliorates influenza pneumonia and protects from morbidity and mortality". We have provided a detailed response below to the comments and suggestions. We believe that the revision has improved the quality of the manuscript. Reviewer comments are in italics.
Response to Reviewer #1
Comment 1: The authors seem to present this approach as innovative, yet this strategy was already considered more than 20 years ago. I therefore think that the authors should include a paragraph in the "introduction" section specifying the novelty that their work brings to the subject. The targeting of STAT3 is however new and original. The results are very impressive, especially if one considers the survival of the animals with this treatment. The use of mice invalidated for different forms of TNF is very relevant and adds much to the demonstration.
Response: We agree with Reviewer 1. We have now included one paragraph (lines 57-64) that summarises some previous studies on therapies targetting both virus and the immune response or targeting TNF to dampen inflammation in the Introduction to provide context. We have also included one paragraph (lines 110-116) to highlight the novelty of our study. The two paragraphs read as follow:
Lines 57-64
“Some previous studies have evaluated the therapeutic potential of adjunctive anti-inflammatory drug interventions in severe respiratory diseases. Corticosteroids in combination with antiviral agents were shown to effectively alleviate the 2009 pandemic H1N1 influenza-associated pneumonia [13]. Similarly, Zheng et al. demonstrated amelioration of lung pathology and improved survival rates in H5N1 IAV-infected mice through co-administration of cyclooxygenase inhibitors and zanamivir [14]. Finally, Shi and colleagues reported that treatment with the anti-TNF drug, etanercept, at 2 h p.i. with H1N1 IAV protected mice from otherwise lethal infection [15].”
Lines 110-116
“Unlike a previous study [15], we have evaluated the efficacy of etanercept, combined with an antiviral drug, influenza pneumonia in a realistic timeframe that closely reflects timeframes when patients present at hospitals. Treating lung inflammation in humans (even before it has developed) from the day of infection is neither feasible nor practical for various logistical reasons. Additionally, targeting the STAT3 signalling pathway with an inhibitor combined with antiviral drug treatment to ameliorate influenza pneumonia is novel and has not been reported previously.”
Comment 2: Overall, the data presented are relevant and conclusive. However, I believe that the qPCR data in Fig. 4 and 6 should be presented differently, as it is not possible to assess the level of cytokine induction in the infected and untreated condition. The authors should change the reference and use an uninfected control for example.
Response: We agree with Reviewer 1 that it is not possible to assess the level of cytokine induction in the infected and untreated condition with the way in which the data is presented. However, and unfortunately, uninfected animals were not included in experiments, the results of which are presented in Figures 4 and 6. In a previous study (https://www.pnas.org/doi/full/10.1073/pnas.2004688117), we found that minimal levels of mRNA transcripts are expressed in lungs of uninfected mice for most of these cytokines and chemokines.
Reviewer #1 – Minor comments
- Abstract and introduction: the first sentences must be modified since influenza is not the only cause of pneumonia.
Response: We agree and sentences have been modified to reflect this point.
In the Abstract, we have inserted “Influenza before “pneumonia” which now reads as “Influenza pneumonia” (Line 16).
In the Introduction, the first and second sentences have been modified as follows: “Pneumonia is a serious complication caused by inflammation of the lungs following infection with diverse viral pathogens, including seasonal and pandemic influenza A viruses (IAVs), that often result in lung pathology, respiratory failure and death [1-4]. There are currently no treatments for viral pneumonia, including influenza pneumonia” (Lines 37-40).
- Line 43: “An over-exuberant immune response…” please find a more appropriate term to qualify the cytokine storm observed during severe influenza pathology.
Response: We have replaced “over-exuberant” with “hyperactive”. (Line 48)
- Line 180: “change “Clinical Disease” for “Clinical Symptoms”.
Response: We have replaced “Clinical Disease” with “Clinical Signs” as only signs but not symptoms are measurable in mice. (Line 211)
- Line 182: “Etanercept mediates anti-inflammatory effects by neutralizing sTNF and…” to my knowledge, Etanercept does not neutralize sTNF but compete with its receptor.
Response: We have modified the sentence which now reads as “Etanercept dampens inflammation by reducing the availability of sTNF to signal via TNFR and through triggering reverse signaling via mTNF [34,35].” (Lines 213-214)
- Line 320: “…dramatic reductions…” as this event is beneficial to the infected host, I think the authors should use another term than "dramatic".
Response: We have replaced “dramatic” with “significant”. (Line 229)
We have also made some additional minor changes to the manuscript to make it read better. These changes are in lines 64-65 and line 89. The material added to the Introduction (lines 57-64) in the revised manuscript was originally in the Discussion section of the original manuscript. That section has now been removed in the revised manuscript to avoid repetition.

Reviewer 2 Report
Below are my views on the article titled, "Therapeutic targeting of inflammation and virus simultaneously ameliorates influenza pneumonia and protects from morbidity and mortality."
In the experimental study, the antiviral oseltamivir and the anti-inflammatory drug etanercept were used to reduce TNF signaling after the onset of clinical symptoms to treat pneumonia in the influenza A virus (IAV) mouse infection model. As a result of the study, it was determined that combined treatment with oseltamivir and an activator of a signal converter and transcription 3 (STAT3) inhibitor effectively reduced clinical disease and lung pathology. According to the results obtained by the authors, it has been claimed that this combined treatment, which is effective in IAV pneumonia, may also be applicable for the treatment of pneumonia caused by SARS-CoV-2. This claim has scientific content that should be taken into account in terms of public health.
In the "Abstract" part of the article, it has been seen that there is self-qualified information about the purpose, scope, context, result, and effect of the study. The abstract has a feature that motivates the reader to read about the content and context of the article. “Keywords” are relevant and sufficient.
In the "Introduction" part, they made an extensive literature review using 27 references on the subject, and in this way, they presented a model that included the relationships between their studies and other research.
In the "Materials and Methods" section, the authors followed a qualitative and quantitative methodology. The submission of additional materials that make the methodology section valuable was welcomed. In the supplementary text, it is detailed how the viral titer was determined by plaque assay for influenza A virus and the cytopathic effect of virus-induced cells in tissue culture plates was evaluated. In addition, histology and microscopic examination of lung pathology, RNA extraction and cDNA production, the qRT-PCR procedure, and the list of primers used in the study are given. Also in the Supplementary Text is Figure S1, which shows that TNF deficiency exacerbates clinical manifestations and lung pathology in IAV-infected mice, and a combined dose of etanercept with standard dose oseltamivir reduced weight loss, clinical scores, lung pathology, and mRNA transcripts for proinflammatory cytokines and chemokines, but not IAV. Figure S2 shows that it does not reduce viral load in WT mice infected with. The methodology of the study has been enriched with the Supplementary text presented with six references.
In the "Material and Method" part, sources 70, 71, and 72 were used after source 27. References should be renumbered in the order they appear in the text and reordered in the "References" section.
In the "Results" section, etanercept monotherapy, combined treatment with etanercept and oseltamivir, daily combined treatment with etanercept and high-dose oseltamivir, and combined treatment with high-dose oseltamivir and S3I-201 on influenza A virus-infected WT and TM mice, weight loss, clinical and histopathological Six figures showing scores and viral load changes are included.
Articles 13, 20, 26, 27, 30, and 34, in which one or more authors are listed, are cited. Since these citations are directly related to the essence of the work, they were not considered inappropriate autocitations.
The article, which presented the results of an experimental study approved by the ethics committee, was evaluated as the product of a study with a guiding feature in the treatment of IAV infections. As we mentioned above, it was found appropriate to publish the article in the journal if the references are numbered according to their order of occurrence in the text and reordered in the "References" section.
Your valuable article reports the results of a hard-working study. In the Materials and Methods section, references 70, 71, and 72 were used after reference number 27. It would be appropriate to number the references in the order they appear in the text and reorder them in the "References" section.
Author Response
We thank Reviewer 2 for their positive and constructive comments on our manuscript, "Therapeutic targeting of inflammation and virus simultaneously ameliorates influenza pneumonia and protects from morbidity and mortality". We have provided a detailed response below to the comments and suggestions. We believe that the revision has improved the quality of the manuscript. Reviewer comments are in italics.
Comment 1. In the "Material and Method" part, sources 70, 71, and 72 were used after source 27. References should be renumbered in the order they appear in the text and reordered in the "References" section.
Response: We agree with Reviewer 2. We have correctly ordered the references in the revised manuscript.
We have also made some additional minor changes to the manuscript to make it read better. These changes are in lines 64-65 and line 89. The material added to the Introduction (lines 57-64) in the revised manuscript was originally in the Discussion section of the original manuscript. That section has now been removed in the revised manuscript to avoid repetition.
